# Causality between Sex Hormones and Bone Mineral Density in Childhood: Age- and Tanner-Stage-Matched Sex Hormone Level May Be an Early Indicator of Pediatric Bone Fragility

**DOI:** 10.3390/biomedicines12061173

**Published:** 2024-05-25

**Authors:** Sung Eun Kim, Seulki Kim, Shin-Hee Kim, Won Kyoung Cho, Kyoung Soon Cho, Min Ho Jung, Moon Bae Ahn

**Affiliations:** 1Department of Pediatrics, College of Medicine, Incheon St. Mary’s Hospital, The Catholic University of Korea, Seoul 06591, Republic of Korea; libaigh2@naver.com (S.E.K.); tigger1018@naver.com (S.-H.K.); 2Department of Pediatrics, College of Medicine, Eunpyeong St. Mary’s Hospital, The Catholic University of Korea, Seoul 06591, Republic of Korea; seulki12633@gmail.com; 3Department of Pediatrics, College of Medicine, St. Vincent’s Hospital, The Catholic University of Korea, Seoul 06591, Republic of Korea; wendy626@catholic.ac.kr; 4Department of Pediatrics, College of Medicine, Bucheon St. Mary’s Hospital, The Catholic University of Korea, Seoul 06591, Republic of Korea; soon926@hanmail.net; 5Department of Pediatrics, College of Medicine, Yeouido St. Mary’s Hospital, The Catholic University of Korea, Seoul 06591, Republic of Korea; jmhpe@catholic.ac.kr; 6Department of Pediatrics, College of Medicine, Seoul St. Mary’s Hospital, The Catholic University of Korea, Seoul 06591, Republic of Korea

**Keywords:** sex hormones, bone mineral density, hypogonadism

## Abstract

This study aimed to investigate the impact of hypogonadism on bone mineral density (BMD) in children and adolescents with chronic diseases to determine the relationship between sex hormones and BMD. This retrospective study included 672 children and adolescents with chronic diseases such as hemato-oncologic, rheumatoid, gastrointestinal, and endocrinologic diseases. The relationship between the sex- and Tanner-stage-matched *Z-scores* for sex hormones and the sex- and age-matched lumbar spine BMD (LSBMD) *Z-scores* was evaluated. Adjustments were made for confounders such as underlying diseases, age at diagnosis, and age- and sex-matched body mass index *Z-scores*. Patients had a mean LSBMD *Z-score* of −0.55 ± 1.31. In the multivariate regression analysis, male testosterone showed a positive association with the LSBMD *Z-score* (*p* < 0.001), whereas female estradiol, luteinizing hormone, and follicular-stimulating hormone showed no significant association with the LSBMD *Z-scores*. In the male group, the testosterone level was associated with LSBMD *Z-scores* > −1.0 (*p* < 0.001), > −2.0 (*p* < 0.001), and > −3.0 (*p* = 0.002), while the estradiol level was associated with LSBMD *Z-scores* > −2.0 (*p* = 0.001) and > −3.0 (*p* = 0.002) in the female group. In conclusion, sex hormones are associated with BMD in children and adolescents with chronic diseases. Therefore, various measures may be necessary to predict future skeletal problems and improve bone health in these patients.

## 1. Introduction

Chronic diseases in childhood, which are often underestimated due to their rarity, cause several medical problems with prolonged morbidity [1]. These diseases are associated with a wide variety of medical fields, including hemato-oncology, rheumatology, gastroenterology, and endocrinology. Patients with these diseases experience several medical problems due to the natural progression of the diseases and also due to the various side-effects that arise from the treatments they receive. In addition, side-effects persist even after the completion of treatments, causing a deterioration in the quality of life of the patients. Bone weakness is a common complication in these patients [2]. Decreased bone mineral density is widely observed in patients with chronic diseases, possibly due to hormonal or metabolic changes, immobility, nutritional deficiency, chronic inflammation, or the use of bone-toxic drugs (for example, steroids and other immunosuppressive agents) during treatments [3]. Several previous studies have shown that children and adolescents are more vulnerable to the side-effects of these drugs and that bone metabolism is more affected in people of these ages [4,5].

In the context of hormonal changes in children with chronic diseases, hypogonadism is a possible consequence of chronic diseases. Gonadal failure is often observed in children with hemato-oncological diseases [6]. These patients receive radiation and gonadotoxic agents as an adjuvant therapy for extensive diseases, which results in the destruction of the reproductive organs [7]. Hypogonadism and pubertal delay are also common when inflammatory bowel diseases (IBDs) such as Crohn’s disease (CD) and ulcerative colitis (UC) are present during childhood [8]. A persistent inflammatory state, steroid treatments, and an altered body composition with cachexia in rheumatic diseases such as juvenile idiopathic arthritis are considered to be risk factors for gonadal dysfunction, which may result in pubertal disorders such as pubertal delay and growth retardation [9]. Various endocrinological diseases such as multiple pituitary hormone deficiency, hypogonadotropic hypogonadism, primary gonadal failure, growth hormone deficiency, and hypothyroidism have been proven to be associated with hypogonadism; however, the underlying mechanisms may vary [10,11].

Among the sex hormones, estrogen is well known for its function in bone metabolism. Estrogen plays a major role in bone turnover in both females and males. Moreover, it also reportedly promotes growth throughout puberty [12]. Compared with estrogen, the role of testosterone in bone metabolism is relatively undefined and its impact on bone health has been deemed to be modest in the past [13]. Recent studies have suggested that testosterone plays an important role in maintaining bone mineral density (BMD) and bone health among males [14]. However, these studies were mostly confined to adults and studies in children are lacking. In addition to these sex hormones, gonadotropins such as luteinizing hormone (LH) and follicular-stimulating hormone (FSH) are sex-related hormones that may contribute to bone health. Although several studies have been conducted to elucidate the cause-and-effect relationship between gonadotropins and bone turnover, this relationship remains unclear [15].

This study aimed to investigate the impact of reproductive hormones on bone metabolism in children and adolescents with chronic diseases by analyzing the roles of sex hormones and gonadotropins in the context of BMD and their relationships. The study was conducted to validate the hypothesis that a deficiency of sex hormones may negatively impact on bone mass in children and adolescents with chronic diseases.

## 2. Materials and Methods

### 2.1. Study Population

This study included patients diagnosed with various chronic diseases. The medical records of 672 children and adolescents aged 10–18 years who were diagnosed with chronic diseases at Seoul St. Mary’s Hospital (a tertiary care center) were retrospectively and cross-sectionally reviewed. Patients with chronic diseases were defined as those who had been diagnosed with certain diseases that required long-term treatments for at least 1 year and who had completed disease-associated treatments at least 1 year prior to the study enrollment. Regular follow-ups, either on an inpatient or outpatient basis, were performed for these patients. Among the eligible patients, those who had undergone an investigation for BMD at least once were included. The enrolled patients were classified into the following four different medical branches according to their disease characteristics: hemato-oncologic, rheumatoid, gastrointestinal, and endocrinologic. Patients who could not be classified into any of the aforementioned categories were designated as ‘unclassified’. The disease category was similar to that used in previous studies, which has been described previously [16,17]. The study was approved by the Institutional Review Board of the Catholic University of Korea (KC23RIS10607) and was conducted in accordance with the tenets of the Declaration of Helsinki. Written informed consent was obtained from all the participants and their parents or guardians.

### 2.2. Disease Category

#### 2.2.1. Hemato-Oncologic Diseases

Patients previously diagnosed with lymphoma, histiocytosis, leukemia, solid organ tumors, or bone-marrow failure were included. The patients were treated with surgery, multiple blood transfusions, multidrug chemotherapy, and/or irradiation, with or without peripheral blood stem-cell transplantation. The treatment plan followed the uniform protocol of the Division of Pediatric Hematology.

#### 2.2.2. Rheumatoid Diseases

Patients previously diagnosed with rheumatoid diseases such as systemic lupus erythematosus, juvenile rheumatoid or idiopathic arthritis, or Sjögren syndrome were included in this category. The patients were treated with steroids, nonsteroidal anti-inflammatory drugs (e.g., naproxen, ibuprofen, and celecoxib), and other antirheumatic drugs (e.g., methotrexate, colchicine, hydroxychloroquine, and biologics). The treatment plan followed the uniform protocol of the Division of Pediatric Rheumatology.

#### 2.2.3. Gastrointestinal Diseases

Patients previously diagnosed with inflammatory bowel disease, including CD and UC, were included in this category. The patients were treated with steroids, 5-aminosalicylic acid, and immune modulators with or without biologics (such as antitumor necrosis factor or anti-integrin). The treatment plan followed the uniform protocol of the Division of Pediatric Gastroenterology and Nutrition.

#### 2.2.4. Endocrine Diseases

Patients previously diagnosed with growth hormone deficiency, idiopathic short stature, hypophosphatemic rickets, idiopathic osteoporosis, or congenital adrenal hyperplasia were included in this category. The patients were treated with hormone replacement therapy according to their hormone levels. The treatment plan followed the uniform protocol of the Division of Pediatric Endocrinology.

### 2.3. Outcome Variables

#### 2.3.1. Baseline Characteristics and Anthropometric Measurements

Clinical data associated with the results of the BMD examinations were collected. Information regarding underlying chronic disease, sex, age at BMD examination, age at chronic disease diagnosis, and anthropometric measurements (height, weight, and BMI) was collected. A Harpenden Stadiometer (Holtain^®^, Crymych, UK) was used to measure height (centimeters, cm) and a weighing scale was used to measure weight (kilograms, kg). BMI was calculated using the Quetelet index (kg/m^2^) and changed to age- and sex-matched *Z-scores* in accordance with the Korean standard growth curve [18]. In addition, clinical data on the sex maturation rate (SMR) at the time of the BMD examinations were collected because sex hormone levels vary depending on the pubertal stage in adolescence [19]. The SMR was recorded using the Tanner scale [20]. Breast development in girls and testicular development in boys were also evaluated.

#### 2.3.2. Hormonal Assays

In the context of reproductive hormones and bone health, laboratory data related to reproductive functions were measured when the patients visited the Pediatric Endocrinology department for regular follow-ups. For accuracy, fasting morning blood samples (approximately between 8 and 10 a.m.) were obtained to determine the sex hormone (testosterone for boys and estradiol for girls) and gonadotropin (LH and FSH) levels [21]. The test was usually performed on the day of the BMD examination. However, for a few patients, samples were collected within 1 month before or after the BMD examination. Sex hormones and gonadotropins were measured as total testosterone (ng/mL), total estradiol (pg/mL), LH (IU/L), and FSH (IU/L) using an Access Immunoassay system (Beckman Coulter, Inc., 250 S. Kraemer Blvd., Brea, CA, USA). This system detects sex hormones and gonadotropins using a paramagnetic particle-based chemiluminescent immunoassay. All values were converted into sex- and Tanner-stage-matched *Z-scores*.

#### 2.3.3. Bone Assessment

To examine the BMD score, the lumbar spine BMD (LSBMD) was measured in the anterior–posterior direction using dual-energy X-ray absorptiometry (Horizon W DXA system^®^, Hologic Corp., Marlborough, MA, USA). The lumbar spine was selected for examination because the guidelines of the International Society for Clinical Densitometry recommend the lumbar spine and total body (less the head) for the evaluation of pediatric BMD [22]. BMD measurements were obtained by radiographers blind to the clinical history of the participants. Areal BMD (g/cm^2^) values were converted into sex- and age-matched *Z-scores* based on the Korean reference data [23]. In addition, a height-adjusted LSBMD *Z-score* was calculated using the reference values of body size adjustments in Korean children and adolescents [24]. Both the LSBMD *Z-score* and height-adjusted LSBMD *Z-score* were additionally classified into three groups with different cut-offs for negative scores (> −1.0, > −2.0, and > −3.0) to further examine the relationship between hormones and BMD. The fracture history of the participants was not reviewed in this study and the term ‘osteoporosis’ was not evaluated. Bone turnover markers, which can be helpful when assessing bone health, were not included in the analysis because laboratory data regarding bone resorption (e.g., the C- or N-terminal telopeptide of type 1 collagen) or bone formation (e.g., bone-specific alkaline phosphatase, procollagen type 1 N propeptide, or osteocalcin) were lacking for the majority of the study population.

### 2.4. Statistical Analysis

All descriptive variables were expressed as the mean ± standard deviation (SD) for continuous variables and as numbers (percentages) for categorical variables. The Shapiro–Wilk test was used to determine the normal distribution of continuous variables. LSBMD *Z-scores* regarding the disease category, age, and Tanner stage were compared using one-way ANOVA or Kruskal–Wallis tests, depending on the normal distribution of variables. Sex hormones and gonadotropins across different ages were compared in the same manner. Univariate and multivariate linear regression analyses were performed to estimate the beta coefficients of the factors associated with the LSBMD *Z-scores*. The calculated sample size for a multiple regression analysis with a confidence level of 95% and a margin of error of 5% was 138 for both sexes and the number of subjects involved in this study sufficiently covered the required sample size. To calculate the odds ratio (OR) of participants presenting three different LSBMD *Z-score* ranges (>−1.0, >−2.0, or >−3.0), a multiple logistic model was used. The results for both unadjusted reproductive hormones and sex- and Tanner-stage-matched reproductive hormones were calculated. During this process, patients with missing records of the Tanner stage at the time of their BMD examination were excluded from the adjustment. Regression analyses were repeated for each disease category in the same manner. Additionally, regression analyses were conducted with height-adjusted LSBMD *Z-scores* for the total study population. A statistical significance was set at *p* ≤ 0.05. SPSS software (version 24.0; IBM Corp., Armonk, NY, USA) was used for all statistical analyses.

## 3. Results

### 3.1. Demographics of the Study Participants

The clinical characteristics of the study population are summarized in Table 1. Among the 672 patients included in the study, approximately half of the study population consisted of males and the mean age of the study participants was 13.6 ± 2.32 years. Among the underlying disease categories, the proportion of patients with hemato-oncologic diseases was the highest (74.4%), followed by rheumatoid (11.1%), gastrointestinal (7.4%), endocrinological (6.2%), and unclassified diseases (4.9%). Among patients with hemato-oncologic diseases, lymphoid or myeloid leukemia/lymphoma accounted for 71.4%, followed by primary bone-marrow failure such as aplastic anemia (14.8%), and solid tumors (8.2%). The most prevalent rheumatoid diseases were systemic lupus erythematosus (60%) and juvenile arthritis (18.2%), either rheumatoid or idiopathic. Gastrointestinal diseases accounted for 6.3% of the cases. Of these patients, the majority (76.2%) were diagnosed with inflammatory bowel diseases such as CD or UC. The overall populations with endocrinological disorders (5.5%) and miscellaneous conditions (5.7%), including anorexia nervosa, epilepsy, hydrocephalus, and nephrotic syndrome, were similar. Patients diagnosed with any syndromic condition with or without chromosomal numerical aberrations such as Turner, Down’s, or Klinefelter syndromes as well as those with primary genetic conditions interfering with sex hormone production were excluded. Children who had previously been diagnosed with primary bone disease or who died during the treatment period were also excluded. The time interval between an endocrinological assessment and a diagnosis of underlying disease was 4.41 ± 3.9 years.

### 3.2. BMD Status of the Study Participants

The mean lumbar spine bone mineral density (LSBMD) *Z-score* of the study participants was −0.55 ± 1.31, and those belonging to the age ranges of 10–12 years and 16–18 years showed the highest and lowest *Z-scores*, respectively (Table 1). The LSBMD *Z-score* of the age 10–12 group was statistically different from the age groups of 12–14 (*p* = 0.002), 14–16 (*p* < 0.001), and 16–18 (*p* = 0.002) years. Approximately half (65.2%) of the entire study population had a low bone mass and those with *Z-scores* < −1.0 and <−2.0 accounted for 39.1% and 12.1% of the study population, respectively. The mean LSBMD *Z-scores* were −0.53 ± 1.29, −0.5 ± 1.47, −0.67 ± 1.25, −0.98 ± 1.28, and −0.41 ± 1.3 for patients with hemato-oncologic, rheumatoid, gastrointestinal, endocrinologic, and unclassified diseases, respectively. A comparison of LSBMD *Z-scores* according to the underlying disease category and Tanner stage is shown in Figure 1. The LSBMD *Z-scores* did not demonstrate any difference among the disease categories. In terms of sexual maturity, there were only two patients at Tanner stage I (prepubertal) and they were both male. The degree of the sexual maturity ratings, except for the female LSBMD *Z-scores* of Tanner II, was significantly higher than those of the LSBMD *Z-scores* of Tanner III (−0.33 ± 1.16 vs. −0.97 ± 1.19; *p* = 0.007) (Figure 1B).

### 3.3. Hormonal Comparison among Different Age Groups

The mean sex hormone *Z-scores* according to the disease category are presented in Table 1. The mean male testosterone *Z-scores* in all disease categories were positive, although the values were close to zero. Contrastingly, female estradiol levels were near to −1.0 in all disease categories. The LH, FSH, and testosterone concentrations in males across different age groups from 10 to 18 years of age are shown in Figure 2A. In the male participants, the highest levels of LH and FSH were 3.29 ± 4.28 and 8.33 ± 11.15 U/L, respectively, both at the age of 14 years. The mean differences in the LH and FSH levels between 10 and 14 years of age were −2.11 U/L (*p* < 0.001) and −5.64 U/L (*p* = 0.006), respectively. Testosterone levels gradually increased with age, showing the highest concentration at 18 years of age (4.62 ± 1.18 ng/mL). The mean difference in testosterone level was greatest at −4.21 ng/mL between the ages of 9 and 16 years (*p* = 0.012). The LH, FSH and estradiol concentrations in females across different age groups from 10 to 18 years of age are shown in Figure 2B. In the female participants, the highest LH and FSH levels were 17.24 ± 34.98 and 30.31 ± 39.89 U/L, which were observed at the ages of 16 and 17 years, respectively. The mean difference in the LH level was greatest at −12.96 U/L between 12 and 16 years of age (*p* = 0.002). On the other hand, FSH levels did not show any significant difference among the age groups. Similar to testosterone, estradiol levels also gradually increased with age, showing the highest concentration at the age of 17 years (111.1 ± 118.6 pg/mL). The mean estradiol concentration at 18 years of age was 84 ± 87.3 pg/mL, which was based on an assessment of five participants. The mean difference in estradiol was greatest at −76.9 pg/mL between 10 and 17 years of age (*p* = 0.001).

### 3.4. Further Analysis of People with BMD ≤ −2.0

A further analysis was conducted with 81 people whose LSBMD results were ≤ −2.0 (Table 2). In general, the characteristics of this group were similar to the total population regarding sex, age, and composition of chronic diseases. The mean height and weight *Z-scores* were −1.62 ± 1.45 and −1.59 ± 1.50, respectively, notably decreased compared with those of the total population. The LSBMD *Z-scores* were most decreased in the ages between 12 and 14. The sex- and Tanner-matched *Z-scores* of testosterone and estradiol were decreased compared with the total study population, regardless of the underlying conditions. The difference was more prominent for the serum estradiol level in females.

### 3.5. Association of BMD with LH, FSH, Testosterone, and Estradiol

A total of 659 subjects with records of their Tanner stage at the time of their BMD examination were included in this analysis. The LSBMD *Z-scores* were linearly correlated with the *Z-scores* of testosterone (*p* = 0.012) and estradiol (*p* = 0.005), and inversely with FSH *Z-scores* (*p* = 0.033) (Figure 3). The value of the correlation coefficient was the highest between LSBMD and estradiol (*r* = 0.138), followed by that between LSBMD and testosterone (*r* = 0.126), and LSBMD and FSH (*r* = −0.083) (Figure 3B–D). There was no direct correlation between the LSBMD *Z-scores* and LH *Z-scores* (Figure 3A). A univariate regression analysis revealed that the male LSBMD *Z-scores* were positively associated with the *Z-scores* for sex- and Tanner-stage-matched testosterone (*β* = 0.9, 95% confidence interval (95% CI) = 0.4–1.39, and *p* < 0.001) and sex- and age-matched body mass index (BMI) (*β* = 0.34, 95% CI = 0.4–1.39, and *p* < 0.001), and negatively associated with age at diagnosis of underlying disease (*β* = −0.06, 95% CI = −0.09–−0.02, and *p* < 0.001) (Table 3). On the other hand, the female LSBMD *Z-scores* were positively associated with the *Z-scores* of sex- and Tanner-stage-matched estradiol (*β* = 0.15, 95% CI = 0.04–0.26, and *p* = 0.009) and sex- and age-matched BMI (*β* = 0.24, 95% CI = 0.16–0.32, and *p* < 0.001), and negatively associated with the sex- and Tanner-stage-matched FSH (*β* = −0.12, 95% CI = −0.22–−0.02, and *p* = 0.025). The sex- and Tanner-stage-matched LH *Z-scores* were linearly and inversely associated with the LSBMD *Z-scores* of male and female participants, respectively; however, these associations were not statistically significant. The sex- and Tanner-stage-matched *Z-scores* of testosterone (β = 1.27, 95% CI = 0.82–1.71, and *p* < 0.001) of the male participants remained positively related to the LSBMD *Z-scores* in the multivariate regression analysis. Contrarily, estradiol in female participants showed no correlation with the LSBMD *Z-scores* in the multivariate study. In the same analyses, the sex- and age-matched BMI *Z-scores* showed an independent linear association with the LSBMD *Z-scores* for both male (*β* = 0.36, 95% CI = 0.28–0.44, and *p* < 0.001) and female (*β* = 0.22, 95% CI = 0.13–0.29, and *p* < 0.001) participants. Regarding the multivariate regression analyses for each disease category (Appendix A), the LSBMD *Z-scores* were associated with the male testosterone *Z-scores* of hemato-oncologic diseases (*p* < 0.001), male LH *Z-scores* of rheumatoid diseases (*p* = 0.049), and female estradiol *Z-scores* of endocrinologic diseases (*p* = 0.006). The multivariate regression analysis conducted for height-adjusted LSBMD *Z-scores* showed a negative correlation with the LH *Z-scores* in males, while both sex hormone *Z-scores* did not achieve a statistical significance (Appendix A).

Logistic regression analyses were performed to compare the effects of each hormone on the LSBMD *Z-scores* when the severity of a low BMD gradually decreased (Table 4). The unadjusted ORs of male LH levels for LSBMD *Z-scores* < −1.0, −2.0, and −3.0 were 33.71 (95% CI = 4–283.86; *p* = 0.001), 200 (95% CI = 3.98–10,050.5; *p* = 0.008), and 5698 (95% CI = 6.47–5,020,000; *p* = 0.012), respectively. On the other hand, female LH levels did not show any association with the LSMBD *Z-scores*. After adjusting for confounders such as underlying diagnosis, age at diagnosis of the underlying disease, and age- and sex-matched BMI *Z-scores*, the OR increased with a decrease in the LSBMD *Z-score*. Unadjusted testosterone had ORs of 2.74 (95% CI = 1.19–6.3; *p* = 0.018), 57.4 (95% CI = 7.86–419.11; *p* < 0.001), and 350 (95% CI = 14.14–8659.2; *p* < 0.001) for LSBMD *Z-scores* > −1.0, −2.0, and −3.0, respectively. After adjusting for the aforementioned confounders, the OR became stronger than when it was unadjusted and increased with a decrease in the LSBMD *Z-score*. Both the unadjusted and adjusted ORs of estradiol only increased between a −2.0 and −3.0 *Z-score* interval of LSBMD. FSH levels showed no significant association with any *Z-score* interval of LSBMD. Regarding the logistic regression analysis for each disease category (Appendix A), male testosterone *Z-scores* were associated with LSBMD *Z-scores* > −1.0, −2.0, and −3.0 for hemato-oncologic diseases and female *Z-scores* were associated with LSBMD *Z-scores* > −2.0 and −3.0 for hemato-oncologic diseases. Sex hormones were not associated with LSBMD *Z-scores* in other disease categories. A logistic regression analysis additionally conducted for a height-adjusted LSBMD *Z-score* showed that the estradiol *Z-score* was a predictive factor for height-adjusted LSBMD *Z-scores* > −1.0, −2.0, and −3.0, whereas the testosterone *Z-score* was not a predictive factor (Appendix A).

## 4. Discussion

This study examined the BMD status of patients with various chronic diseases across several medical sectors. The majority of the study population had hemato-oncologic diseases and the mean sex- and age-matched LSBMD *Z-score* was lower than that of the normal population in South Korea. More than half of the study population had an LSBMD *Z-score* of < 0, and the score tended to decrease with an increase in age. In a subgroup analysis, failure to thrive and a decrease in serum estradiol levels were noteworthy for LSBMD *Z-scores* ≤ −2.0. The LSBMD *Z-score* was found to be closely related to BMI in the multivariate regression analysis. Among males, the association between the testosterone level and LSBMD *Z-score* was significant. On the other hand, the estradiol level in females was found to be irrelevant to the LSBMD *Z-score*. In addition, logistic regression analyses showed that low testosterone and LH levels were risk factors for low LSBMD *Z-scores* in males. The regression analyses also demonstrated that a low estradiol level was a risk factor among females with an LSBMD *Z-score* of <−2.0.

In contrast to primary osteoporosis, which occurs due to intrinsic skeletal defects, secondary osteoporosis results from chronic systemic illnesses either due to the effects of the disease process or due to the treatments that the patients undergo. Practical guidelines for assessments of bone health among children and adolescents with the aforementioned chronic diseases have been established by the Korean Society of Pediatric Endocrinology [25]. However, risk assessments of low BMD must be differentiated based on disease characteristics because the disease progression and treatment are essentially different. In the context of hemato-oncologic diseases, aggressive treatments that children undergo primarily aggravates BMD. Long-term treatments with chemotherapy and immunosuppressive agents are a strong risk factor for low BMD in childhood cancer survivors [26,27,28]. Other important treatments such as hematopoietic stem-cell transplantation or irradiation for high-risk childhood leukemia have also been suggested to be potential risk factors for a low bone mass [29,30]. The negative impact of glucocorticoids and osteoimmune interactions on bone mass has been reported in previous studies on patients with systemic lupus erythematosus and juvenile idiopathic arthritis [31,32,33]. Osteotoxic effects of drugs such as glucocorticoids and infliximab have also been observed in patients with IBD [34,35]. Compared with a normal population, patients with IBD reportedly experience vertebral fractures more frequently and with comparatively greater severity, particularly among those with CD [36]. A deficiency of hormones such as a growth hormone or thyroid hormone has been suggested to be a risk factor for low BMD in patients with growth hormone deficiency or hypothyroidism [16,37,38,39]. In this study, the majority of the patients had low BMD, regardless of the type of disease they had. Although the underlying mechanisms for a weakened bone density may differ, our study demonstrated that children and adolescents with chronic diseases remain vulnerable to bone problems even after completing treatments. This highlights the importance of regular BMD examinations and measures to maintain bone health. Regarding the differences in bone mineral density among the disease categories, the lowest LSBMD *Z-score* was observed in patients with endocrinological diseases, but the difference between the groups was not statistically significant.

Hypogonadism is a potential risk factor for low bone mass in chronic diseases, and the relationship between hypogonadism and childhood cancer survivors has been relatively clearly defined. A previous study reported that childhood cancer survivors with secondary hypogonadism induced by cranial irradiation had a lower BMD than those with eugonadism [40]. Gonadal dysfunction has been suggested to be a prominent risk factor for low BMD; an irradiation treatment is likely to play a major role in this process, thereby affecting any stage of the hypothalamic–pituitary–gonadal axis [41]. Although some studies have suggested that hypogonadism is a possible factor for low bone mass in other disease categories such as gastrointestinal and rheumatoid diseases, the evidence is scarce and further research is needed in this area [42,43]. In this study, we observed low *Z-scores* for female estradiol in different disease categories and hypogonadism was even more prominent in female patients with LSBMD *Z-scores* ≤ −2.0. Contrastingly, the *Z-scores* of male testosterone showed slightly better results in comparison to a normal population, but the mean scores decreased for male patients with an LSBMD *Z-score* ≤ −2.0. In terms of sex hormone levels among different age groups, a gradual increasing pattern of male testosterone and female estrogen across ages was observed, which was compatible with normal pubertal development. In the case of gonadotropins, however, this gradual increasing pattern was only observed in females, while LH and FSH levels in males reached a peak at the age of 14. This may imply pubertal males with chronic disease are more vulnerable to secondary hypogonadism, but further studies are necessary to clarify the causal association. In the regression analysis, this study demonstrated a linear correlation between low sex hormone levels and low LSBMD *Z-scores* in both sexes. However, the clinical importance of testosterone and estradiol differed because the LSBMD *Z-scores* showed more sensitivity to testosterone levels in males than to estradiol levels in females.

Among the several hormones involved in bone metabolism in children, estrogen is the key hormone that controls bone maturation during childhood. Before the progression of puberty, bone maturation observed in females is thought to be an effect of estrogen because the estrogen level in females is higher than that in males [44]. Estrogen also plays a key role in pubertal males. In male patients with estrogen resistance or aromatase deficiency, the fusion of growth plates is delayed and, consequently, the height continues to increase [45]. The positive relationship observed between female estradiol levels and LSBMD *Z-scores* in our study was in concordance with the aforementioned studies. However, it is noteworthy that the estradiol level was not associated with the LSBMD *Z-scores* in the multivariate analysis. Considering the fact that the estradiol level was not associated with LSBMD *Z-scores* > −1.0, the impact of estrogen is assumed to be evident in female patients with more advanced bone mineral loss. Testosterone is a relatively novel hormone that has recently attracted considerable attention. Although the function of testosterone in bone remodeling and its impact on BMD have been well determined in the past century [46], studies on testosterone levels and BMD in adult patients have not shown significant results [47,48,49]. However, a few studies have reported that testosterone levels are closely associated with pubertal growth and bone maturation in male adolescents [50,51,52]. In the same context, our findings also demonstrate that testosterone might be essential for bone maturation in males during puberty, and its clinical significance is not inferior to that of estradiol in females. As these sex hormones dramatically change during growth, our study is noteworthy because the hormones adjusted with the Tanner stage. As far as we know, there is no other study that has conducted an analysis of the correlation between BMD and Tanner-stage-matched sex hormones. Tanner-stage-matched sex hormones are more suitable to compare with age-matched sex hormones because hormonal changes in puberty are closely related to the Tanner stage [20].

The relationship between reproductive hormones other than sex hormones and bones was also investigated in this study. This study focused on two gonadotropins (LH and FSH), which regulate the sex hormones. Several previous observational studies have reported an association between increased serum FSH levels and bone loss; however, these studies primarily focused on adults and most of them targeted postmenopausal women [53,54,55,56]. No study has clarified this relationship in children or adolescents. In addition, no human intervention study has detected a direct cause-and-effect relationship between FSH levels and bone turnover. The relationship becomes even more confusing in adolescents. Our study also showed an irrelevant relationship between FSH and LSBMD *Z-scores* in both sexes. In the case of LH, we observed a positive relationship between the LH increment and a favorable LSBMD *Z-score* (scores > −1.0, −2.0, and −3.0) in males, implying that LH may directly participate in the bone mineral loss seen in children and adolescents with chronic diseases. In the same context, the distinctive pattern of the male LH levels across different ages may provide some insights to understand the relationship between LH and bone health. Although several previous studies have consistently reported that the effect of LH on bone is minimal [57,58,59], large prospective studies focusing on the relationship between LH and BMD, especially in adolescents, are lacking.

This study had a few limitations that need consideration. First, this was a retrospective cross-sectional study. Therefore, the longitudinal effects of chronic disease morbidity on BMD could not be defined. Second, data on male estrogen and female testosterone levels were not available. Investigating the cause-and-effect relationship between these two hormones and BMD would have deepened our understanding of the significant findings observed in this study. Third, this study only measured the BMD of the lumbar spine. The BMD score of the total body (less the head) is clinically important when interpreting BMD studies for children [22]. However, it was not measured due to technical issues in the corresponding hospital. We acknowledge that this is an important limitation that confines the data analysis in a pediatric study. Fourth, information on vertebral fractures was not included in the data analyses. An evaluation of bone density does not solely depend on the BMD score, and vertebral fractures are a clinical prerequisite to assess osteoporosis in children. Finally, other disease-specific factors that may have had an impact on the BMD results were not included. Periods of treatment with glucocorticoid or other immunosuppressive agents, different disease stages of each chronic disease, and nutritional factors of the patients are factors that strongly impact BMD in these chronic diseases. In the case of nutritional factors, our consideration of the BMI in the regression analysis would have complemented the limitation to some extent as the BMI is a useful parameter that reflects the nutritional state. An individual analysis of each chronic disease was additionally conducted in this study, but we did not observe any meaningful results from rheumatoid, gastrointestinal and endocrinologic diseases, mainly due to the small sample size of the corresponding disease categories. As this study focused on the general impact of sex hormones on BMD across various diseases, future studies analyzing its impact on specific diseases are necessary. Despite these limitations, our study also had its strength in the context that sex- and Tanner-stage-matched adjustments were made for sex hormone levels, which more accurately reflected the status of reproductive function.

## 5. Conclusions

Reduced levels of sex hormones may be an early sign of bone fragility in children and adolescents with chronic conditions. Hence, it may be necessary to conduct routine assessments of sex- and Tanner-matched testosterone and estradiol levels from the onset of disease and throughout the treatment period to prevent secondary osteoporosis. Future studies that complement the aforementioned limitations of our study are warranted to elucidate the mechanisms underlying the complex relationship between reproductive hormones and bone density.

## Figures and Tables

**Figure 1 biomedicines-12-01173-f001:**
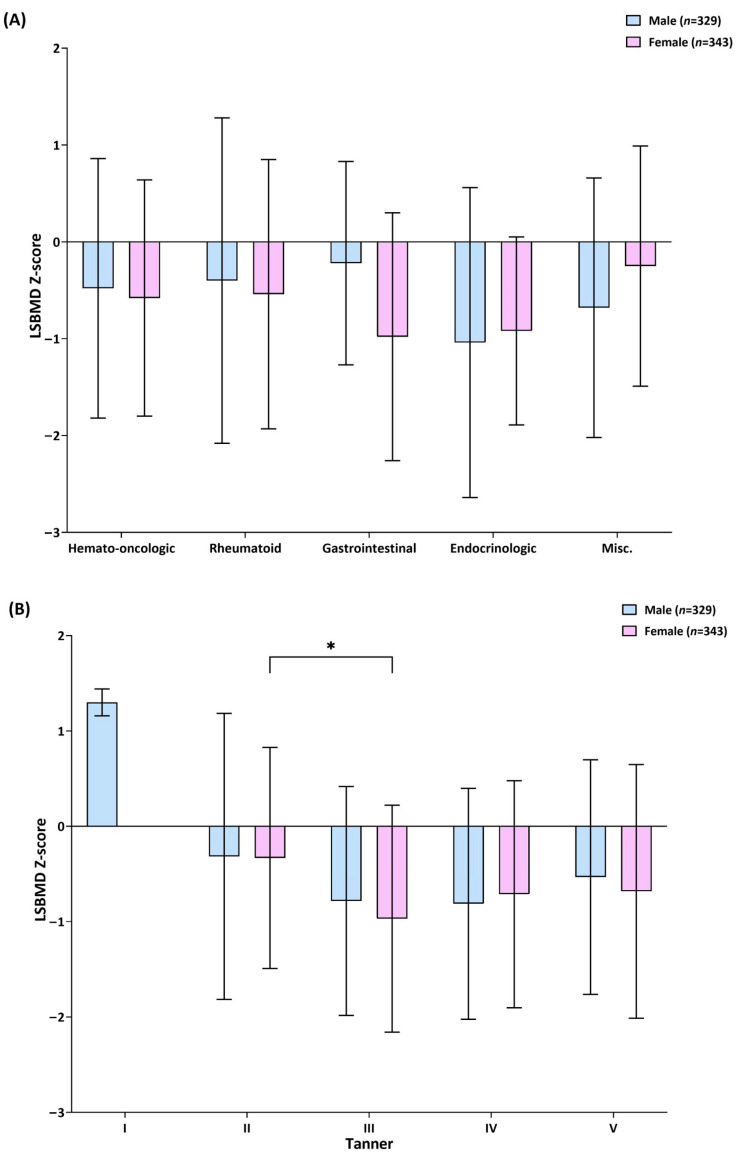
Bar graphs representing the male and female lumbar spine bone mineral density (LSBMD) *Z-scores*. (**A**) Bar graphs based on disease category (hemato-oncologic, rheumatoid, gastrointestinal, endocrinologic, and misc. (unclassified)). (**B**) Bar graphs based on Tanner stage. Whiskers indicate standard deviations. The asterisk (*) indicates when it is significantly differed.

**Figure 2 biomedicines-12-01173-f002:**
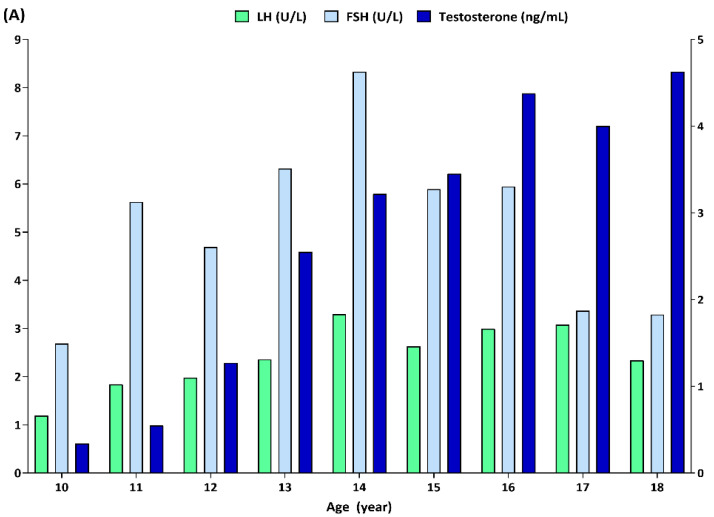
Histograms demonstrate luteinizing hormone (LH), follicular-stimulating hormone (FSH), testosterone, and estradiol for male (**A**) and female (**B**) subjects from age 10 to 18. The values for LH and FSH correspond with the left y-axis, while those of testosterone and estradiol correspond with the right y-axis. Units for hormonal assays are described.

**Figure 3 biomedicines-12-01173-f003:**
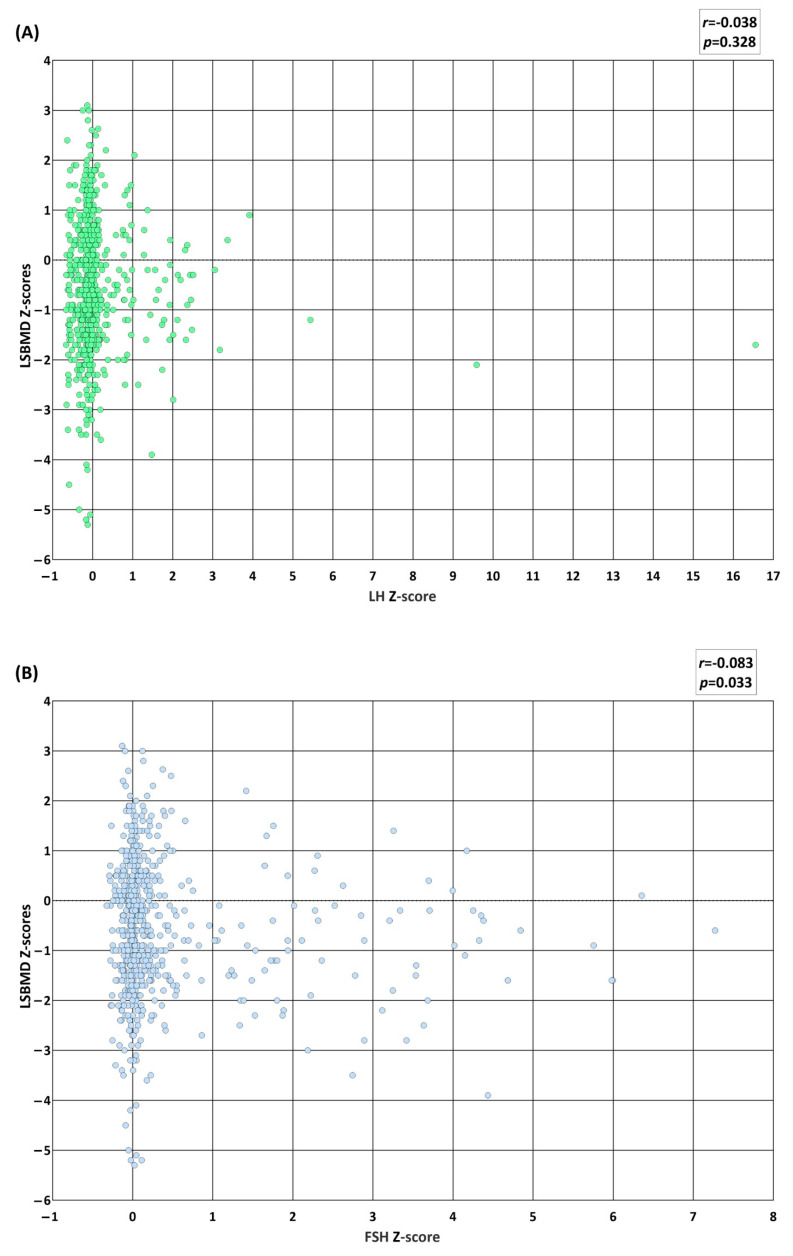
Scatter plots representing the relationship between sex- and age-matched lumbar spine bone mineral density (LSBMD) *Z-scores* and sex- and Tanner-matched gonadotropin or sex hormone *Z-scores*. LSBMD *Z-scores* (y-axis) are displayed against (**A**) luteinizing hormone (LH) *Z-scores*, (**B**) follicular-stimulating hormone (FSH) *Z-scores*, (**C**) male testosterone *Z-scores*, and (**D**) female estradiol *Z-scores*.

**Table 1 biomedicines-12-01173-t001:** Clinical description of the study subjects.

		Total (*n* = 672)
**Male, *n* (%)**		329 (48.9)
**Age, years**		13.6 ± 2.32
**Anthropometry, *Z-score***		
Height		−0.52 ± 1.36
Weight		−0.35 ± 1.55
BMI		−0.13 ± 1.60
**Tanner stage, *n* (%)**		
I		2 (0.3)
II		261 (38.8)
III		86 (12.8)
IV		93 (13.8)
V		217 (32.3)
**Underlying condition, *n* (%)**		
Hemato-oncologic		500 (74.4)
Rheumatoid		55 (8.1)
Gastrointestinal		42 (6.3)
Endocrinologic		37 (5.5)
Unclassified		38 (5.7)
**Age at diagnosis, years**		9.24 ± 4.6
**Lumbar spine BMD, *Z-score***		−0.55 ± 1.31
Age, years		
10–12		−0.25 ± 1.27
12–14		−0.68 ± 1.36
14–16		−0.67 ± 1.19
16–18		−0.71 ± 1.37
Low bone mass, *n* (%)		
≤0		438 (65.2)
≤−1.0		263 (39.1)
≤−2.0		81 (12.1)
≤−3.0		23 (3.4)
**Sex hormone, *Z-score***	**Testosterone (male)**	**Estradiol (female)**
Hemato-oncologic	0.16 ± 0.28	−0.81 ± 1.29
Rheumatoid	0.10 ± 0.31	−1.11 ± 1.05
Gastrointestinal	0.31 ± 0.32	−1.48 ± 1.07
Endocrinologic	0.14 ± 0.32	−1.00 ± 0.92
Unclassified	0.25 ± 0.45	−1.56 ± 0.88

All values are expressed as mean ± standard deviation unless mentioned. BMI: body mass index; BMD: bone mineral density.

**Table 2 biomedicines-12-01173-t002:** Clinical description of the study subjects for lumbar spine BMD ≤ −2.0.

		Total (*n* = 81)
**Male, *n* (%)**		43 (53.1)
**Age, years**		14.4 ± 2.17
**Anthropometry, *Z-score***		
Height		−1.62 ± 1.45
Weight		−1.59 ± 1.50
BMI		−0.97 ± 1.54
**Tanner stage, *n* (%)**		
II		19 (23.5)
III		14 (17.3)
IV		15 (18.5)
V		31 (38.3)
**Underlying condition, *n* (%)**		
Hemato-oncologic		57 (70.4)
Rheumatoid		9 (11.1)
Gastrointestinal		6 (7.4)
Endocrinologic		5 (6.2)
Unclassified		4 (4.9)
Age at diagnosis, year		10.26 ± 4.72
**Lumbar spine BMD, *Z-score***		−2.77 ± 0.84
Age, years		
10–12		−2.98 ± 1.20
12–14		−2.99 ± 1.06
14–16		−2.59 ± 0.43
16–18		−2.66 ± 0.70
**Sex hormone, *Z-score***	**Testosterone (male)**	**Estradiol (female)**
Hemato-oncologic	−0.01 ± 0.25	−1.46 ± 0.92
Rheumatoid	−0.04 ± 0.18	−1.53 ± 0.90
Gastrointestinal	−0.18 ± 0.00	−1.74 ± 0.95
Endocrinologic	−0.17 ± 0.21	−1.71 ± 1.10
Unclassified	0.06 ± 0.18	−1.63 ± 0.00

All values are expressed as mean ± standard deviation unless mentioned. BMI: body mass index; BMD: bone mineral density.

**Table 3 biomedicines-12-01173-t003:** Univariate and multivariate regression analyses of factors associated with the lumbar spine bone mineral density of male and female subjects.

	LSBMD *Z-scores*							
	Male				Female			
	Univariate		Multivariate		Univariate		Multivariate	
	β	*p*	β	*p*	β	*p*	β	*p*
**Age at diagnosis**	**−0.06**	** *<0.001* **	**−0.06**	** *<0.001* **	−0.02	*0.187*		
**^†^ BMI**	**0.34**	** *<0.001* **	**0.36**	** *<0.001* **	**0.24**	** *<0.001* **	**0.22**	** *<0.001* **
**^‡^ LH**	0.68	*0.075*			−0.06	*0.241*		
**^‡^ FSH**	0.151	*0.492*			**−0.12**	** *0.025* **	−0.08	*0.109*
**^‡^ Testosterone**	**0.9**	** *<0.001* **	**1.27**	** *<0.001* **				
**^‡^ Estradiol**					**0.15**	** *0.009* **	0.07	*0.203*

^†^ Sex- and age-matched *Z-scores*. ^‡^ Sex- and Tanner-stage-matched *Z-scores*. BMI: body mass index; FSH: follicular-stimulating hormone; LH: luteinizing hormone, LSBMD: lumbar spine bone mineral density.

**Table 4 biomedicines-12-01173-t004:** Logistic regression models demonstrating the effect of luteinizing hormone, follicular-stimulating hormone, testosterone, and estradiol on lumbar spine bone mineral density.

	LSBMD *Z-scores*											
	>−1.0				>−2.0				>−3.0			
	Male		Female		Male		Female		Male		Female	
	OR	*p*	OR	*p*	OR	*p*	OR	*p*	OR	*p*	OR	*p*
**^†^ LH**												
**Unadjusted**	**33.71**	** *0.001* **	0.92	*0.345*	**200**	** *0.008* **	0.96	*0.739*	**5698**	** *0.012* **	1.22	*0.671*
**^‡^ Adjusted**	**63.51**	** *<0.001* **	0.93	*0.417*	**134.81**	** *0.022* **	0.97	*0.779*	**3124.35**	** *0.034* **	1.23	*0.682*
**^†^ FSH**												
**Unadjusted**	1.58	*0.226*	0.94	*0.477*	4.06	*0.089*	0.86	*0.162*	14.5	*0.073*	0.79	*0.236*
**^‡^ Adjusted**	1.8	*0.142*	0.96	*0.869*	4.25	*0.093*	0.87	*0.224*	17.15	*0.081*	0.79	*0.248*
**^†^ Testosterone**												
**Unadjusted**	**2.74**	** *0.018* **			**57.4**	** *<0.001* **			**350**	** *<0.001* **		
**^‡^ Adjusted**	**5.19**	** *<0.001* **			**60.68**	** *<0.001* **			**243.33**	** *0.002* **		
**^†^ Estradiol**												
**Unadjusted**			1.58	*0.08*			**1.82**	** *<0.001* **			**5.07**	** *0.011* **
**^‡^ Adjusted**			1.12	*0.308*			**1.97**	** *0.001* **			**13.53**	** *0.002* **

^†^ Sex- and Tanner-stage-matched *Z-scores*. ^‡^ Adjusted for underlying disease, age at diagnosis of underlying disease, and age- and sex-matched body mass index *Z-scores*. FSH: follicular-stimulating hormone; LH: luteinizing hormone; LSBMD: lumbar spine bone mineral density; OR: odds ratio.

## Data Availability

All the data analyzed in this are not publicly available for the privacy of the research participants but are available from the corresponding author (M.A.B) upon reasonable request.

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
