# Peer review of "Causality between Sex Hormones and Bone Mineral Density in Childhood: Age- and Tanner-Stage-Matched Sex Hormone Level May Be an Early Indicator of Pediatric Bone Fragility"

_biomedicines, 2024, doi:10.3390/biomedicines12061173_

Round 1

Reviewer 1 Report

Comments and Suggestions for Authors

Summary:

This manuscript reported a retrospective cross-sectional study that aimed to investigate the impact of hypogonadism on bone mineral density (BMD) in children and adolescents with chronic diseases. This study mainly assessed the relationship between sex hormones and BMD. The authors recruited 672 children and adolescents with chronic diseases, such as haemato-oncologic, rheumatoid, gastrointestinal, and endocrinologic diseases. Relationship between sex- and Tanner stage-matched z-scores for sex hormones and sex- and age-matched lumbar spine BMD (LSBMD) z-scores were evaluated. The authors corrected the confounders, such as underlying diseases, age at diagnosis, and age- and sex-matched body mass index Z-scores. The results showed that patients had a mean LSBMD Z-score of −0.55±1.31. The multivariate regression analysis showed that male testosterone having a positive association with LSBMD Z-score (p<0.001), whereas female estradiol, luteinizing hormone, and follicular stimulating hormone having no significant association with LSBMD Z-scores. In the male group, the testosterone level was associated with LSBMD Z-scores >-1.0 (p<0.001), >-2.0 (p<0.001), and >-3.0 (p=0.002), while estradiol level was associated with LSBMD Z-scores >-2.0 (p=0.001) and >-3.0 (p=0.002) in the female group. The authors concluded that sex hormones are associated with BMD in children and adolescents with chronic diseases. They suggested various measures may be used to predict future skeletal problems and improve bone health in these patients.

Comments:

This retrospective study showed that sex hormones are associated with BMD in children and adolescents with chronic diseases. This study focused on the secondary osteoporosis. This study recruited a group of patients with different kinds of disorders. Most disorders of these subjects may directly influence BMD through different mechanisms, e.g., inflammation, malnutrition, impaired bone formation, accelerated bone loss, etc. The authors attempted to analyze the effects of sexual hormones on BMD. However, they did not take into these effects in the analysis. The conclusion may change if other more factors were included. Since different disorders may have their own impacts, it may be simple to individually analyze the association among each category of patients. Other more issues listed following needed to be addressed before reaching a solid conclusion. This manuscript in current status may not reach the standard for being considered for publication in Biomedicines.

Other comments:

1.     z-scores should be changes to Z-scores

2.     Lines 101-129 These categories of disorders may undergo glucocorticoid treatment (line, 106-107, 113, 119) or have related disturbance due to disease themselves (lines, 126-129). Therefore, the effects of the level of glucocorticoid on BMD may be assessed.

3.     Lines 147-155. The blood samples were collected approximately between 8-10 AM) were obtained to determine sex hormones (testosterone for boys and estradiol for girls) and gonadotropin (LH, FSH) levels. These samples may also be used for bone markers measurements. These data may also be helpful to assess the mechanisms of secondary osteoporosis.

4.     There are many different kinds of disorders in different stages. The authors may need to check the status of disorders, i.e., in active status, stable status, under current treatment, how long the period after last treatment, especially the hormone replacement or glucocorticoid treatment. How about the duration of disorders of these subjects? How about the severity of disorders? Did the authors assess these parameters by classification? Will they influence the BMD?

5.     Lines 190 Inflammatory bowel diseases, such as CD or UC, They may influence the BMD through nutritional factors instead of hormone only. The authors may separate these subjects and reanalyze these data again to exclude the confounding factors and reach a solid conclusion

6.     Line 199 Table 1 The effect of age seems greater in the teenagers as compared with children. Endocrine functions change dramatically during growing. For example, menstruation in girls may influence the level of sexual hormone. These factors may also need check in this study.

7.     Line 199 Table 1. Subjected with Z-scores ≤−2.0 81(12.1%) and ≤−3.0 23 (3.4%) may be important clues for this study of secondary osteoporosis. These small group may need deep analysis to find out something important.

8.     The body height and stature may influence BMD. It was mentioned in lines Therefore, BMD score of total body less head is clinically important in interpreting BMD studies for children it was not measured due to technical issues in the corresponding hospital (Lines 396-397). Thus the conclusion may change if reanalysis using such BMD data. The authors may need to present inclusion criteria and exclusion criteria for controlling confounding factors.

Comments on the Quality of English Language

The manuscript was well written. 

Reviewer 2 Report

Comments and Suggestions for Authors

Comments on the Quality of English Language

English language requires minor editing.

Reviewer 3 Report

Comments and Suggestions for Authors

The study assessed the association between testosterone and estrogen and lumbar spine BMD z-scores in children with different diseases. I have the following comments:

I suggest adding a hypothesis statement to the end of the introduction section.

Line 156: Why did you not assess BMD at the proximal femur? This is also an important fracture site for osteoporotic fracture.

Statistical analyses: Please justify your sample size calculation for your multiple regressions. This is usually based on the number of predictor variables in your multiple regression.

Lines 210-214: Please ensure that you have included a description of the statistics for these comparisons in the statistics section.

Lines 223-227: It is not necessary to repeat numerical results here that are already presented in the table (this is redundant).

Lines 227-242: Again, please ensure numerical results are not presented twice (i.e., in text and then again in a table or figure). Also, please ensure that the statistics for these comparisons between different groups is described in the statistics section.

Line 249: Delete the word “Causal” from the subheading here because correlations by rule cannot demonstrate causality.

Line 256: Please clarify that the confidence intervals presented are 95% confidence intervals.

Figure 3: In all these graphs, should lumbar spine BMD z-score be on the y-axis, as this is the dependent variable?

Round 2

Reviewer 1 Report

Comments and Suggestions for Authors

The revised manuscript well addressed the comments. Some  comments were listed as limitation of this study. 

Comments on the Quality of English Language

The manuscript was well written. 

Author Response

Thank you for reviewing my paper and providing important comments!

Reviewer 3 Report

Comments and Suggestions for Authors

Thanks for responding to my comments

Author Response

(The authors gave the same response as above.)
